# Effects of Target of Rapamycin and Phosphatidylinositol 3-Kinase Inhibitors and Other Autophagy-Related Supplements on Life Span in *y w* Male *Drosophila melanogaster*

**DOI:** 10.3390/ijms252111504

**Published:** 2024-10-26

**Authors:** Aaron A. Bearden, Emily M. Stewart, Candace C. Casher, Meredith A. Shaddix, Amber C. Nobles, Robin J. Mockett

**Affiliations:** Department of Biomedical Sciences, University of South Alabama, Mobile, AL 36688-0002, USA; aaron.bearden@health.southalabama.edu (A.A.B.); ems1221@jagmail.southalabama.edu (E.M.S.); ccc2121@jagmail.southalabama.edu (C.C.C.); mas1929@jagmail.southalabama.edu (M.A.S.); anobles@uabmc.edu (A.C.N.)

**Keywords:** life span, aging, *Drosophila*, rapamycin, TOR inhibition

## Abstract

Various dietary supplements have been shown to extend the life span of *Drosophila melanogaster*, including several that promote autophagy, such as rapamycin and spermidine. The goal of the study presented here was to test numerous additional potential anti-aging supplements, primarily inhibitors of the target of rapamycin (TOR) and/or phosphatidylinositol 3-kinase (PI3K). Using a single, comparatively long-lived *y w* test strain, screening was performed in male flies supplemented either throughout adulthood or, in a few cases, beginning in middle or late adult life, with concentrations spanning 4–6 orders of magnitude in most cases. Supplementation with PP242 and deferiprone, an iron chelator, beginning in late adult life had no positive effect on life span. Lifelong supplementation with Ku-0063794, LY294002, PX-866-17OH, Torin2 and WYE-28 had no effect at any dose. Rapamycin, spermidine and wortmannin all had significant life-shortening effects at the highest doses tested. AZD8055, PI-103 hydrochloride and WYE-132 yielded slight beneficial effects at 1–2 doses, but only 100 nM AZD8055 was confirmed to have a minor (1.3%) effect in a replicate experiment, which was encompassed by other control groups within the same study. These compounds had no effect on fly fecundity (egg laying) or fertility (development of progeny to adulthood), but equivalent high doses of rapamycin abolished fertility. The solvent DMSO had no significant effect on life span at the concentrations used to solubilize most compounds in the fly medium, but it drastically curtailed both survival and fertility at higher concentrations. 2-Hydroxypropyl-β-cyclodextrin also failed to extend the life span when provided throughout adulthood or beginning in mid-adult life. Collectively, the results suggest that inhibition of the TOR/PI3K pathway and autophagy through dietary intervention is not a straightforward anti-aging strategy in *Drosophila* and that further extension of life is difficult in comparatively long-lived flies.

## 1. Introduction

*Drosophila melanogaster* is widely used as a model organism to test the effects of dietary supplements on aging and longevity [1]. Life extension has been reported at one or more concentrations for the following supplements, among many others: rapamycin (sirolimus), which promotes autophagy as an inhibitor of the target of rapamycin (TOR) signal transduction pathway downstream from phosphatidylinositol 3-kinase (PI3K) [2,3]; the rapamycin analog (rapalog) everolimus and other kinase inhibitors [4]; and spermidine, a natural polyamine that inhibits oxidative stress and also promotes autophagy [5,6].

The PI3K/AKT/TOR pathway is of particular interest because mutations in all components of the pathway are common in various solid tumors [7] and because rapamycin itself extends life and diminishes the incidence of cancer in numerous strains of mice [8,9,10]. The rapalogs everolimus and temsirolimus are used to treat tuberous sclerosis complex (TSC) [11,12], which arises from TSC1/2 mutations that up-regulate the pathway, and some human cancers [13,14,15]. Rapamycin is also used off-label by some healthy adults for the potential extension of health span and even life span [16]. Nonetheless, numerous authors express varying levels of concern about known side effects for both rapamycin and rapalogs [9,11,12,13,14,15], and the limited effectiveness of rapalogs against cancer has led to their being superseded by other modes of treatment in some cases [15,17] and to the investigation of alternative means of inhibiting TOR activity [13,18]. In the context of aging, low or intermittent dosing of rapamycin is an option to minimize side effects [9,16]. In female *w*^Dah^ *Drosophila*, treatment for 15–30 days in early adult life was sufficient to extend the median adult life span from 72 to 78–80 days [3].

The TOR protein is a serine/threonine kinase that functions as the catalytic subunit in two multiprotein complexes (TORC1 and TORC2) that were first identified from spontaneous mutations conferring rapamycin resistance in *Saccharomyces cerevisiae* [19]. TORC1 is activated by nutrients [9,20] and by growth factors acting through PI3K via AKT inhibition of TSC1/2 [18,21] or through mitogen-activated protein kinase (MAPK) [22]. It phosphorylates ribosomal protein S6 kinase (S6K) [23,24] and eukaryotic initiation factor 4E-binding protein 1 (4E-BP1) [24,25] to promote translation and growth in nutrient-rich conditions [21]. TORC2 is likewise activated by increases in nutrient levels and by growth factors via PI3K, but it is additionally activated by decreases in nutrient levels and by various forms of stress [26]. It phosphorylates at least 26 targets and has more diverse functions, including (i) activation of AKT and PKC, which promote TORC1 signaling, (ii) activation of SGK, which also promotes TORC1 and additionally up-regulates ion channels, glucose and amino acid carriers, (iii) actin cytoskeletal remodeling and (iv) either positive or negative regulation of autophagy [26]. Both TORC1 and TORC2 also have roles in promoting lipid biosynthesis [27].

It has been proposed that inhibition of TORC1 is beneficial for life span, but inhibition of TORC2 is detrimental [9,28]. Rapamycin acutely inhibits TORC1, which is thought to give rise to its life-extending effect, but prolonged exposure to rapamycin also interrupts the assembly of TORC2 to a cell-type-specific extent [29], leading to side effects such as insulin resistance [28]. The best cost–benefit ratio might therefore be achieved using low doses, intermittent or transient exposure to rapamycin, or compounds that are more highly selective for TORC1 [9]. However, the regulation of either complex in isolation is not straightforward because of feedback loops whereby TORC1 inhibits the upstream insulin receptor substrate-1 and rapamycin relieves this inhibition [30,31], and because TORC2 activates TORC1 via phosphorylation of AKT, while TORC1 negatively regulates TORC2 via phosphorylation of its Rictor component by S6K [30,32]. Furthermore, although TORC1 action against S6K is highly sensitive to inhibition by rapamycin [28], in mammalian cells rapamycin only partly or temporarily suppresses the phosphorylation of 4E-BP1 and the activation of autophagy [33,34].

Alternative inhibitors of the TOR pathway that vary in their selectivity for TORC1, TORC2 and PI3K might therefore be of value to obtain or improve upon the benefits of rapamycin while avoiding its side effects. Second-generation TOR kinase inhibitors block both complexes but not the feedback activation of PI3K, whereas dual kinase inhibitors block both TOR and PI3K [18]. Although neither category has yielded favorable outcomes for cancer in initial trials compared with rapalogs [13], preliminary results are somewhat more promising, but not always consistent, for longevity in *Drosophila*. The TOR-selective inhibitor 1 (Torin1) caused a substantial but quite variable increase in once-mated Dahomey female longevity from a short (mean 22 d) baseline [35]. Torin2, which has comparable selectivity for mTOR over PI3K in cellular but not biochemical assays and ~10-fold greater bioavailability than Torin1 [36], with an effective dose ~1/1000 that of rapamycin against neuroblastoma cell lines [37], nonetheless increased the Canton S male median life span by only 4% at the lowest levels of a 0.5–10 µM dose range and had no beneficial effect in females with a longer (median 55 d) baseline [38]. AZD8055 promotes autophagy and inhibits both TOR complexes with at least 1000× selectivity vs. PI3K and other kinases [39]. PP242 also inhibits both complexes with high selectivity relative to PI3K and other kinases [40]. In combination with the senolytic navitoclax (but not separately), either AZD8055 or PP242 increased survival times from a 6-day baseline in female *w*^1118^ flies exposed to hydrogen peroxide and sensitized senescent cells to navitoclax more effectively than rapamycin [41]. In contrast, in male flies of the same strain, lifelong supplementation with PP242 increased mean survival from ~46 to ~57 d, comparable to the effect of rapamycin [42]. LY294002, which was initially identified as a PI3K inhibitor [43] but subsequently found to inhibit TOR [44] and numerous unrelated kinases as well [45], was reported to extend the life span at 5 µM in Canton S flies of both sexes from a 51–54 d baseline [46], but it had no effect at 0.5–4.5 mM in Oregon R males under mass-screening conditions that yielded 20–30 d average life spans [4]. Similarly, wortmannin was found first to inhibit PI3K [47,48] and then TOR [44] and to extend the life span slightly from a 33–35 d baseline at 0.5 µM in Canton S males but not females [46], while it failed to extend life in the mass screen of Oregon R males at 0.5–4.5 mM concentrations [4]. It was separately reported to have no significant effect at 5 nM, while it increased Canton S male and decreased female median life spans at 5 µM and benefited both sexes in combination with either rapamycin (at 5 µM) or the NF-κB inhibitor pyrrolidine dithiocarbamate (at 5 nM) [49].

The primary goal of the experiments reported here was to retest the longevity effects of rapamycin, spermidine, Torin2, AZD8055, LY294002 and wortmannin supplementation beginning in early adult life using a *y w* strain and environmental conditions associated with comparatively long life, as well as to test several additional kinase inhibitors and other potential life-extending supplements. Additional TORC1/TORC2 inhibitors included Ku-0063794 [50], WYE-28 [51] and WYE-132 [52], all of which are even more highly selective than PP242. Conversely, the wortmannin derivative PX-866 shows high selectivity for PI3K over TOR [53]. Its more potent primary biological metabolite PX-866-17OH [54,55] was tested, along with a soluble hydrochloride salt of PI-103, which inhibits PI3K class IA isoforms with >100× higher potency than LY294002 [56,57] while also inhibiting TORC1, TORC2 and DNA-PK [57]. Biochemical IC_50_ concentrations of the tested inhibitors of TOR and PI3K, where available, are summarized in Table 1 [58,59] while noting that cellular EC_50_ values and relative selectivity are substantially different for Torin2 [36] and are likely to differ for other compounds as well. If selective inhibition of any one kinase or, alternatively, a balance of inhibition of multiple kinases to various extents is optimal for life extension, then testing a wide range of compounds was predicted to maximize the chance of identifying an ideal agent and concentration. A pilot test of supplementation beginning in late adult life was first performed for PP242 and the iron chelator deferiprone (DFP), which induces mitophagy and decreases the production of reactive oxygen species [60]. DFP at a 163 µM dose partially rescued the life-shortening effect of frataxin knockdown in a *y w* background, although it had no notable effect on the control life span [61]. Supplementation was also initiated in early or mid-adult life for 2-hydroxypropyl-β-cyclodextrin (2-HP-β-CD), which inhibits AKT/TOR phosphorylation and facilitates the initiation of autophagy, although in HepG2 cells, it also blocks the later stages of autophagy and leads to apoptosis [62]. Lastly, the solvent dimethylsulfoxide (DMSO), which was used to solubilize many of the TOR/PI3K inhibitors, has itself been reported to shorten the life span of *Drosophila* at 0.5% but not at 0.1% [63] and to extend the life of *Caenorhabditis elegans* (in the presence of 5-fluorouracil) at 0.5–2.0% [64]. It was therefore tested for its own effects on life span that could mask or amplify the effects of the dissolved supplements. Given that the degree of oral uptake and bioavailability of the tested compounds in flies is not known, the supplements were tested over the widest practical range of concentrations, with minimum doses near to or below their IC_50_ and maximum doses 3–5 orders of magnitude higher in most cases. The concentrations of rapamycin and spermidine were chosen to match those used in previous studies [2,3,5]. The experiments were performed in male flies, with the intention of repeating the tests in females and testing for potential trade-offs related to fertility for any compound at any dose yielding consistent life extension.

## 2. Results

### 2.1. Longevity Effects of DMSO, PP242 and Deferiprone Supplementation Late in Adult Life

An initial test of longevity effects was performed in *y w* male flies on a Torula yeast-based medium, with DMSO, PP242 and DFP supplementation beginning at 64 days of adult age (Figure 1). Mortality prior to this time was ≤10% in all groups. A statistically significant 17% decrease in the remaining survival time was observed for water-supplemented (H_2_O) in comparison with regular control flies (Figure 1, Table 2), with elevated mortality coinciding with insufficient drying of vials on a few occasions. The *p* values for all group comparisons are shown in Table 2. None of the supplemented flies surpassed the survival times of flies on the regular control medium. In comparison with the H_2_O controls, DMSO at final concentrations of 0.0002–0.2% had no significant effect on survival, but 2% DMSO cut the remaining survival time by 75%, from 21.0 to 4.4 days (Figure 1A). PP242 at the highest dose (649 nM) increased survival time by 10% relative to H_2_O controls, while the lowest doses decreased it by 9–14% (Figure 1B). Deferiprone at 7.19–719 µM did not affect survivorship vs. the H_2_O control; at 71.9–719 nM, it increased survival up to essentially the same length as that in the regular control group (Figure 1C).

### 2.2. Longevity Effects of Spermidine and TOR/PI3K Inhibitors—Screening Study

Spermidine and the remaining TOR/PI3K inhibitors listed in Table 1 were tested in six nearly concurrent experiments commenced over an 8-day interval, with lifelong supplementation in continuous darkness beginning 1–2 days after the collection of 0–1-day-old adults (8 days for LY294002). Independent regular and H_2_O control groups were included in each experiment, except for Torin2 and PI-103 hydrochloride, which included only an H_2_O control group. Survivorship curves for all supplements are shown in Figure 2. Mean adult life spans and logrank test results are shown in Table 3 for each experiment, consisting of 1–3 supplements and the corresponding control groups. No significant differences in survivorship were observed among the 11 control groups in the six experiments (*p* = 0.097). In the first experiment, spermidine (dissolved in water) was associated with a 10–11% reduction in life span at the highest dose (10 mM) and no effect at lower doses (10 µM–1 mM). DMSO again had no effect at final concentrations of 0.02–0.2%, corresponding to the highest concentrations for dissolved TOR/PI3K inhibitors. WYE-28 had no effect at concentrations ranging from 10 pM to 1 µM. In the second experiment, WYE-132 was associated with a 5–6% increase in mean life span at 10 pM, the lowest dose tested, but no significant effect at 100 pM–1 µM. LY294002 had no effect at any dose ranging from 100 nM to 10 µM. In the third experiment, AZD8055 extended life by 4–7% at 10 µM and 100 nM while having no effect at 1 µM or 100 pM–10 nM. Rapamycin at high concentrations (100–400 µM) caused a major dose-dependent decrease in life span, ranging from 26 to 45%. The rapamycin groups were also significantly shorter-lived (*p* < 0.0005) than flies supplemented with equivalent or near-equivalent concentrations of DMSO (400, 200 and 100 µM rapamycin vs. 0.2, 0.1 and 0.04% DMSO, respectively). In the fourth experiment, lower doses of rapamycin (100 nM–10 µM) and Ku-0063794 (1 nM–10 µM) also had no effect. Wortmannin had no effect at 1 nM–1 µM, but at 10 µM, it caused a catastrophic 69–70% decrease in mean survival. In the fifth experiment, PX-866-17OH (1 nM–10 µM) did not alter the life span. In the final experiment, Torin2 (100 pM–10 µM) had no effect, while PI-103 hydrochloride (10 µM) increased the mean life span by 4% relative to the H_2_O control group and had no effect at lower doses (100 pM–1 µM).

### 2.3. Longevity Effects of TOR/PI3K Inhibitors—Confirmation Study

An additional experiment was performed under 12 h light/12 h dark conditions for compounds at doses that yielded significant increases in life span in the screening study. Since, in the cases of AZD8055 and PI 103-HCl, the highest concentration (10 µM) increased the life span, here, additional 50 µM treatment groups were added. Rapamycin was also tested at 10 and 50 µM, and a control group containing an identical concentration of DMSO was prepared for each treatment group (Figure 3, Table 4). The variation among control groups was somewhat greater than before, with 0.2% DMSO yielding the longest life span and 0.04% DMSO yielding the shortest life span of any group. AZD8055 (100 nM) was the only treatment that extended life in relation to both regular and H_2_O controls, albeit by only 1.3% vs. regular controls. It also extended life vs. the corresponding 0.0004% DMSO control, but not 0.2% DMSO (Figure 3A).

### 2.4. The Fecundity and Fertility Effects of TOR/PI3K Inhibitors in the Confirmation Study

Early adult fecundity (egg laying) and fertility (number of eggs developing to adulthood) were determined over the first four days of adult life for the same treatment groups as in the life span confirmation study, except that 200 µM rapamycin and a corresponding 0.8% DMSO control group were substituted for the regular and H_2_O controls. Rapamycin at concentrations of 50–200 µM impaired egg laying (Figure 4A) and, at 10–200 µM, totally abolished the development of progeny to adulthood (Figure 4B). All groups conformed to the model assumptions of a normal distribution and equal variance for fecundity, and all except the rapamycin groups conformed for fertility. One-way analysis of variance (ANOVA) showed a significant overall difference in fecundity (*p* = 0.008). Pairwise comparisons between DMSO control and supplementation groups showed marginal decreases (*p* < 0.050) that were not below the Bonferroni-corrected threshold (*p* < 0.006) for 200 µM rapamycin (*p* = 0.007), 50 µM rapamycin (*p* = 0.042) and 10 µM AZD8055 (*p* = 0.046), and no differences for the other groups. For fertility, one-way ANOVA showed no differences among the groups, excluding rapamycin (*p* = 0.27). At a Bonferroni-corrected threshold of *p* < 0.017 for three comparisons, Mann–Whitney U tests revealed significantly lower fertility in the 10 and 50 µM rapamycin groups (*p* = 0.014), but not for 200 µM rapamycin vs. 0.8% DMSO (*p* = 0.046), which itself almost abolished development to adulthood.

### 2.5. Longevity Effect of 2-Hydroxypropyl-β-cyclodextrin (2-HP-β-CD)

The final compound tested was 2-HP-β-CD (Figure 5, Table 5). In an initial experiment, supplementation at concentrations of 0.14–0.56 g/L (corresponding to 96–384 µM based on an average molecular weight of 1460 g/mol) on the standard medium beginning at 50 days of adult age had a positive but nonsignificant effect (+7%) on remaining survival time (Figure 5A). In a replicate experiment, late-life supplementation diminished the survival time by up to 10% at 0.56 g/L, with no significant effect at 0.14 g/L and a borderline significant decrease at 2.80 g/L (Figure 5B). Lifelong supplementation on the standard medium had no effect at all (Figure 5C). On an alternate medium containing more concentrated yeast and sugar but lacking cornmeal, there was an increase in mean life span of up to 5% that was not significant after correction for multiple comparisons (Figure 5D).

## 3. Discussion

The results of this study show that none of the tested TOR and PI3K inhibitors, supplemented over a wide dose range throughout adult life, showed any consistent life-extending effects in male flies of the *y w* test strain, with the minor exception of 100 nM AZD8055, which increased the mean life span by only 1.3% vs. the regular control group in the replicate experiment. In contrast to past reports, rapamycin, wortmannin and the polyamine spermidine all had life-shortening effects at the highest doses. In addition, deferiperone, provided only in late adult life, had no effect, and 2-HP-β-CD, administered either throughout adulthood or beginning in mid-adult life, had no reproducible beneficial effect on longevity. Rapamycin at lower doses did not affect the life span or fecundity, but it completely suppressed the development of eggs to adulthood, in contrast to other TOR/PI3K inhibitors at comparable doses. The results of follow-up experiments examining the effects of rapamycin on longevity and fertility in multiple strains and on multiple diets will be presented and discussed in a separate manuscript.

Several of the supplements in this study (Ku-0063794, PI-103 hydrochloride, PX-866-17OH, WYE-28, WYE-132 and 2-HP-β-CD) have not been tested previously for longevity effects in *Drosophila* or, for AZD8055, have only been tested in the presence of a strong exogenous stressor [41]. For supplements that were tested previously, the current findings confirm the absence of a beneficial effect of DFP in control flies [61]. Life extension was reported for rapamycin [2,3,42,46], LY294002 [46], Torin2 [38], wortmannin [46], PP242 [42] and spermidine [5], although for rapamycin, LY294002 and wortmannin, some investigators reported no change or even adverse effects [4,65]. In general, where differences were observed, plausible explanations include the use of male vs. female flies, genetic differences among fly strains and differences in the supplement dose, food composition or other aspects of the environment. For LY294002, low but not high doses were reported to extend life, albeit in different strains under strikingly dissimilar housing conditions [4,46]. The present study introduces a third strain and shows no effect in *y w* males at the low dose (5 µM) that was beneficial in Canton S males and females [46]. Torin2 was reported to have a slight (4%) positive effect only for 2/4 concentrations in male and 0/4 concentrations in female flies [38], which is not drastically divergent from the 0 ± 1% effect in males in the current study that includes the same concentration range. Again, for wortmannin, the positive effect previously reported for the male mean life span was only 4% at 0.5 µM [46], which does not differ drastically from the 0 ± 2% effect at the same dose in the current study. For higher doses (0.5–4.5 mM), wortmannin was screened and not included in a list of compounds with beneficial effects [4], but it was not clear whether it had neutral or adverse effects. The severe toxicity of 10 µM wortmannin in older flies in the present study is consistent with cell killing by wortmannin in the 2.3–58.4 µM concentration range (levels vastly in excess of its IC_50_ for TOR and PI3K) [48], which suggests that millimolar concentrations would have an even greater detrimental effect. In the case of PP242, the supplement was only administered beginning in late adult life, so the observed life shortening is not directly at odds with the beneficial effect of lifelong supplementation with higher doses reported in *w*^1118^ males [42]. Given that only the lowest doses studied here had an adverse effect and considering the wide variation between control groups in the initial study, the finding of a reduction in life span by PP242 should be treated as very preliminary. Nonetheless, there was no sign of life extension for PP242 and, after replicate experiments were performed, no life extension by lifelong supplementation with PI-103 hydrochloride, which is comparable to PP242 in its IC_50_ values for TORC1 and TORC2 (Table 1). For spermidine, published results show a large beneficial effect in *w*^1118^ females at doses of 10 µM–1 mM and neutral or adverse effects in replicate experiments at 10 mM, whereas in males, only 10 µM spermidine increased the mean adult life span from ~43 to ~50 days, with no effect observed for 100 µM, 1 mM or 10 mM spermidine [5]. In the current study, the lower three of these concentrations had no effect in *y w* males, while 10 mM spermidine was mildly detrimental. Thus, the relative trends for different doses are preserved, with lower doses being more beneficial than higher doses, but in the longer-lived *y w* strain, the overall effect of spermidine is shifted in a negative direction. The variation between life spans in the published and present reports is also comparable in magnitude to the variation among replicate cohorts within each study, and it highlights the importance of repeated testing before reaching firm conclusions about the longevity effects of any treatment in flies.

A limitation of this and some other studies of dietary supplements in *Drosophila* is the lack of a direct demonstration of the uptake of the compounds that were tested. Consequently, the absence of an effect on life span might mean that the supplements are ineffective, or it might result from low bioavailability due to degradation during storage or food preparation, metabolism in the food medium or gastrointestinal system of the flies or an absence of transport across the gut lining. Here, at least, the stability of most of the TOR and PI3K inhibitors and spermidine was demonstrated after the prolonged storage of frozen aliquots and by heating in acid to mimic conditions during food preparation. These results and the toxicity observed at the highest doses of wortmannin, rapamycin and spermidine provide indirect evidence that the absence of effects on life span at other doses or for the other inhibitors is not simply due to low bioavailability. It should also be noted that AZD8055, Ku-0063794, rapamycin, Torin2, WYE-28 and WYE-132 were all supplemented at doses up to at least 1000× higher than their IC_50_ for TORC1, and wortmannin was supplemented at up to >1000× its IC_50_ for PI3K; therefore, a bioavailability of 0.1% of the concentration added to the food should have been sufficient to inhibit the respective kinases. On the other hand, in the case of spermidine, supplements were provided to flies and mice at millimolar concentrations, whereas in isolated cells, effects were observed at nanomolar concentrations [5]. Therefore, an uptake of even 0.1% cannot be assumed. A conclusion that the dietary administration of these compounds confers no benefit to *y w* male flies in this laboratory environment is thus well supported; a conclusion that the compounds themselves are without benefit (for instance, if injection in *Drosophila* were feasible) is only conjectural.

As opposed to low bioavailability, the possibility of excess bioavailability through overdosing should also be considered. While rapamycin had no effect on longevity at 10 and 50 µM, it completely abolished egg development at these concentrations, showing, at least in this case, that the absence of an effect on life span was not simply due to low uptake of the supplement. Based on the molecular weights of the inhibitors in Table 1, an average weight of ~0.7 mg for *y w* male flies in this laboratory and an average food consumption of 2.5 µL/day for *Drosophila* on a comparable diet [66], 10 µM supplementation corresponds to at least 11 mg/kg/day, which would translate to a daily intake of 0.77 g/day for a 70 kg human. For comparison, short-term human trials of rapamycin have used much lower doses, such as 1 mg/day [67]. Therefore, the toxicity of rapamycin and wortmannin at the highest doses used here does not mean they are unsafe for intake at lower levels. Rather, they show that when consumption was pushed to a limit to overcome the potential (and probably real) problem of low bioavailability, the eventual outcome was detrimental rather than beneficial.

2-HP-β-CD was tested in part because it has been shown to diminish the accumulation of lipofuscin [68], an age pigment that may be both a cause and consequence of impaired autophagy [69]. Additionally, the depletion of cellular cholesterol by a related compound, methyl-β-cyclodextrin, suppresses TORC1 activity [20], and 2-HP-β-CD partially rescues the life-shortening effect of the Niemann–Pick type C1 mutation (*npc1*^-/-^) in a mouse model while lowering elevated cholesterol levels in multiple organs [70]. Doses of 500 and 2000 mg/kg/day have no toxicological effects in dogs [71], so equivalent doses of 0.14 and 0.56 g/L (based on 2.5 µL/day food consumption by 0.7 mg flies) were used, along with a 5× higher dose of 2.80 g/L. All doses were well tolerated, but none had a significant positive effect on life span when supplemented beginning in either early or middle adult life. Although the stability of the compound was not checked, it was heated only briefly to ~60 °C during food preparation, whereas thermodegradation of cyclodextrins typically occurs only at >250 °C [72]. The absence of a beneficial effect on life span is consistent with 2-HP-β-CD having no effect on cholesterol content in *npc1*^+/+^ mice, and it provides no evidence for a reversible accumulation of lipofuscin or cholesterol imbalance as a cause of death in the *y w* fly strain. Efforts to quantify lipofuscin in the organs of *y w* flies on the standard diet of this laboratory are in progress.

Finally, the fecundity and fertility study yielded two noteworthy findings. First, rapamycin completely suppressed the development of embryos to adulthood at concentrations where AZD8055 and PI-103 hydrochloride had no effect, even though rapamycin is intermediate between the other kinases in its potency against TORC1. These results might reflect either lower toxicity or faster inactivation of the other inhibitors in comparison with rapamycin. Alternatively, as a natural compound, rapamycin might have other cellular targets that have not yet been identified. Second, the solvent DMSO also nearly completely abolished development at 0.8%, consistent with its severe, life-shortening effect at a 2% final concentration. Although lower concentrations of DMSO used to dissolve TOR/PI3K inhibitors in this study did not significantly affect fertility or longevity (400 µM rapamycin was provided in a final concentration of 0.2% DMSO, 200 µM rapamycin was in 0.1% DMSO, and all other supplements had ≤0.05% DMSO), the potential effects of DMSO as a solvent should be considered in longevity studies. Notably, Spindler and colleagues [4] also provided supplements in a final 0.2% DMSO concentration for their mass-screening experiments and reported average life spans of 20–30 days. Although they adjusted other conditions to achieve the shorter life spans for rapid screening, they noted the need for future experiments with longer control life spans; for such experiments, a control group with no DMSO would exclude an effect of the solvent overlaid on the effects of the supplements.

Although insufficient dosing or bioavailability cannot be excluded as possible reasons for the absence of beneficial effects of the TOR/PI3K inhibitors and other autophagy-related supplements in this study, a straightforward explanation is that this pathway and process are not the limiting factor in the aging and death of male flies of this lineage and dietary regimen. Consistent with these findings, genetic interventions in TOR signaling and autophagy that had been reported by others to extend the life spans of *Drosophila* had no beneficial effect in *y w* (or *w*^1118^) flies of either sex in this laboratory [73]. Although the inhibition of TOR/PI3K or the enhancement of autophagy can be beneficial in some fly strains or environmental conditions, dietary interventions intended to regulate aging by this mechanism do not appear to be universally effective methods for life extension in *Drosophila*.

## 4. Materials and Methods

### 4.1. Fly Strain and Media

The *y w* strain used in this study was generated in the laboratory of W.C. Orr (Southern Methodist University, Dallas, TX, USA) [74] and has been used in previous studies in this laboratory [73]. Fly stocks were maintained and experiments were performed on the previously described medium [73], except for one experiment with 2-HP-β-CD that used a high-yeast medium prepared by boiling 10.0 g/L agar in deionized water, then adding 100 g/L Torula yeast and 50.0 g/L sucrose, boiling again, cooling to ~60 °C, and then adding 3.00 g/L methyl-4-hydroxybenzoate dissolved in 30 mL of ethanol as a mold inhibitor and 3.00 mL/L propionic acid to inhibit bacterial growth.

### 4.2. Supplements

Rapamycin was purchased from LC Laboratories (Woburn, MA, USA). Spermidine and 3-hydroxy-1,2-dimethyl-4(1*H*)-pyridone (deferiprone) were from Sigma Aldrich (St. Louis, MO, USA). The compounds AZD8055, Ku-0063794, LY294002, PI-103 hydrochloride, PP242, PX-866-17OH, Torin2, wortmannin, WYE-28 and WYE-132 were from Chemdea (Ridgewood, NJ, USA). Dimethylsulfoxide (DMSO), ethanol and (2-hydroxypropyl)-β-cyclodextrin (2-HP-β-CD) were from Fisher Scientific (Waltham, MA, USA). Supplements other than spermidine and 2-HP-β-CD were dissolved in DMSO and stored immediately in 25 mM aliquots at −20 °C in a nondefrosting freezer, except for rapamycin (200 mM), PP242 (32.4 mM), WYE-28 (2.5 mM), WYE-132 (2.5 mM) and deferiprone (35.9 mM in water). Spermidine was dissolved in deionized water, filter-sterilized and stored in 1 M aliquots at −20 °C for no more than one month; however, aliquots of TOR/PI3K inhibitors were stored for up to 1 month for the experiments shown in Figure 4, 2–3 months for Table 2 and Table 3 and up to 1 year for Table 4. 2-HP-β-CD was dissolved in deionized water, aliquots of other supplements were thawed, and all supplements were then serially diluted in deionized water just prior to mixing into freshly prepared food (1% *v*/*v* 2-HP-β-CD, 2% *v*/*v* for all others) at ~60 °C, immediately before or after dispensing the medium into individual vials. H_2_O control vials were prepared with an equivalent volume of water, and in most experiments, regular control vials were also prepared without added water. The medium was then allowed to solidify and provided to flies 1–3 days after preparation.

To assess the stability of the supplements under conditions approximating those of the current study, Norman Arundel (Chemdea) performed ultra-performance liquid chromatography (UPLC) with ultraviolet detection using aliquots of the TOR/PI3K inhibitors listed in Table 3 stored in DMSO at −20 °C for 0, 30 and 90 days. Separate aliquots were heated at 65 °C for 30 min in 0.33% propionic acid and 0.033% phosphoric acid, corresponding to the acid and heat exposures during preparation of the standard fly medium. Concentrations ranged from 2.5 to 200 mM during storage and were lowered to 1.25–18 mM by dilution in DMSO before separation on a Waters (Milford, MA, USA) HSS 1.8u column (2.1 × 50 mm) on a Shimadzu Nexera UPLC instrument equipped with an SPD-M40 photodiode array detector and LCMS2020 spectrometer (Shimadzu, Kyoto, Japan). Gradient elution was performed with the alternation between 99% of 0.1% formic acid in water and 80% acetonitrile/20% methanol. In all cases, concentrations ranged from 91 to 106% of the 0-day baseline, and no decomposition products or change in retention time was detected. Spermidine was stored as a 1 M solution in deuterated water at −20 °C for 0, 30 and 90 days or heated in acids at 65 °C for 30 min and detected with AdvanceCore Bruker (Ettlingen, Germany) ^1^H NMR at 400 MHz. There was no noticeable change in NMR peaks due to storage or heating in acid. Functionally, rapamycin stored in DMSO at −20 °C for 95 days maintained the same ability as at 1 day to fully suppress the development of embryos to adulthood at concentrations of 10–50 µM in the standard medium.

### 4.3. Life Span

Male flies were isolated in four groups of 25/vial (25 × 95 mm) (nine groups for regular controls in Figure 1, Table 2) under light carbon dioxide anesthesia 0–1 day after eclosion, excluding immature individuals and any with visible signs of injury, sickness or genetic deformity. They were maintained at 25 ± 1 °C and kept in continuous darkness owing to the light sensitivity of some supplements, except the TOR/PI3K inhibitor confirmation study and experiments with 2-HP-β-CD were performed on a 12 h light/12 h dark cycle. Fresh vials were provided, and deaths or escapes were recorded every 1–2 d, with supplementation beginning at age 64 d for the initial test of DMSO, PP242 and deferiprone. All other supplements were introduced 2 d after collection except LY294002 (8 d), PI-103 hydrochloride (1 d), Torin2 (1 d) and 2-HP-β-CD (either 1 d or 50 d).

### 4.4. Fertility

Groups of three female and three male flies were collected 0–19 h after eclosion and provided with fresh vials containing media with supplements daily for four successive days at 25 ± 1 °C on a 12 h light/12 h dark cycle. Eggs laid were counted when flies were transferred, and adults that eclosed fully within 14 days were also counted.

### 4.5. Statistics

Logrank tests of survivorship were performed using SYSTAT 12 software for the time periods when supplements were provided. For each inhibitor, a test was performed comparing all doses with both regular and H_2_O control groups, where applicable; in cases where the comparison reached a significance threshold of *p* < 0.050 using the method of Mantel–Haenszel, comparisons were performed for each dose vs. both controls. For doses reaching the threshold of *p* < 0.050, paired comparisons were made for each control group separately. For the initial late-life supplementation study, where a difference between control groups was detected, only the H_2_O control was included in the initial comparison, but supplement doses showing a significant difference vs. the H_2_O control group were additionally compared pairwise with the regular control group. In the screening study, to minimize the likelihood of false negative results, and owing to the complexity of the design, with two control and multiple supplementation groups, no adjustment was made for multiple comparisons. Instead, any dose of a compound reaching the threshold of *p* < 0.050 vs. both controls in the screening study was retested in the confirmation study. In the latter case, groups showing significant effects vs. both regular and H_2_O controls were also compared with a control group treated with an identical concentration of the DMSO solvent. For the 2-HP-β-CD study, survival times for all doses were compared with that of the H_2_O control group in each experiment, and pairwise comparisons were made where the comparison reached a significance threshold of *p* < 0.050. Given that only one control group was used, a significance threshold of *p* < 0.017 was set based on Bonferroni correction for three comparisons within each experiment.

For the fertility study, total yields of embryos and subsequently eclosed adults over four days were compared by one-way analysis of variance after using Kolmogorov–Smirnov tests to validate the model assumption of a normal distribution (at a significance threshold of *p* < 0.004 based on Bonferroni correction for 14 multiple comparisons) and Levene’s tests for equal variance. Where significant effects were observed, paired *t*-tests were then performed for individual groups vs. control groups with equivalent volumes of the DMSO solvent (with a significance threshold of *p* < 0.006 after Bonferroni correction for nine comparisons). For groups that did not conform to the model assumptions, Mann-Whitney U tests were used for the paired comparisons.

## Figures and Tables

**Figure 1 ijms-25-11504-f001:**
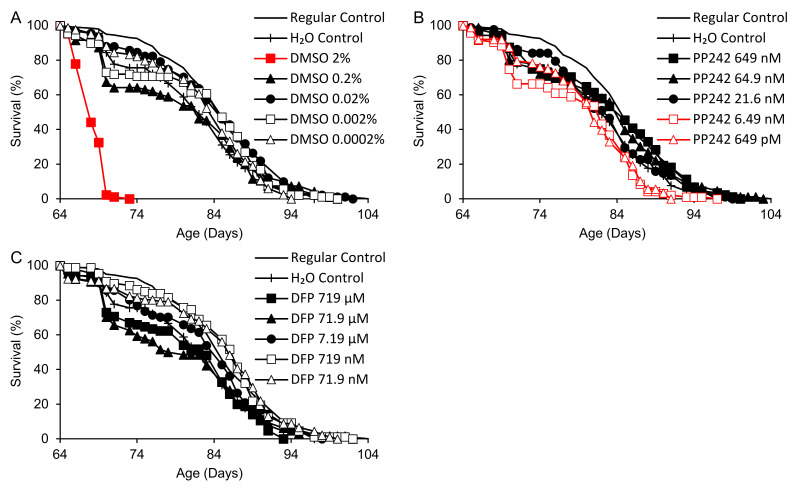
Survival times of flies supplemented with DMSO, PP242 or DFP beginning in late adult life (64 days). (**A**) DMSO. (**B**) PP242. (**C**) DFP. Red: life shortening vs. both regular and H_2_O controls (*p* < 0.050 based on logrank tests).

**Figure 2 ijms-25-11504-f002:**
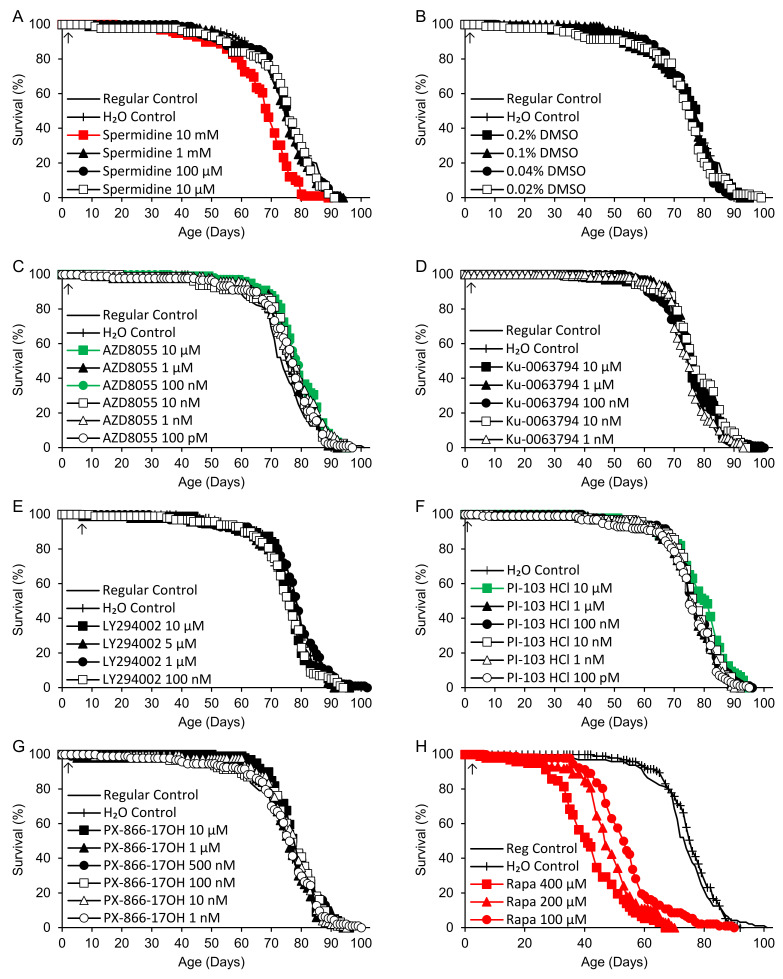
Life spans of flies supplemented with (**A**) spermidine, (**B**) DMSO or TOR/PI3K inhibitors: (**C**) AZD8055, (**D**) Ku-0063794, (**E**) LY294002, (**F**) PI-103 hydrochloride (HCl), (**G**) PX-866-17OH, (**H**) rapamycin (Rapa) high dose, (**I**) Rapa low dose, (**J**) Torin2, (**K**) wortmannin (Wort), (**L**) WYE-28 and (**M**) WYE-132. Supplements were provided beginning in early adult life (arrows). Green indicates life extension and red indicates life shortening vs. both regular and H_2_O controls, or vs. H_2_O controls alone for PI-103 hydrochloride (*p* < 0.050 based on logrank tests).

**Figure 3 ijms-25-11504-f003:**
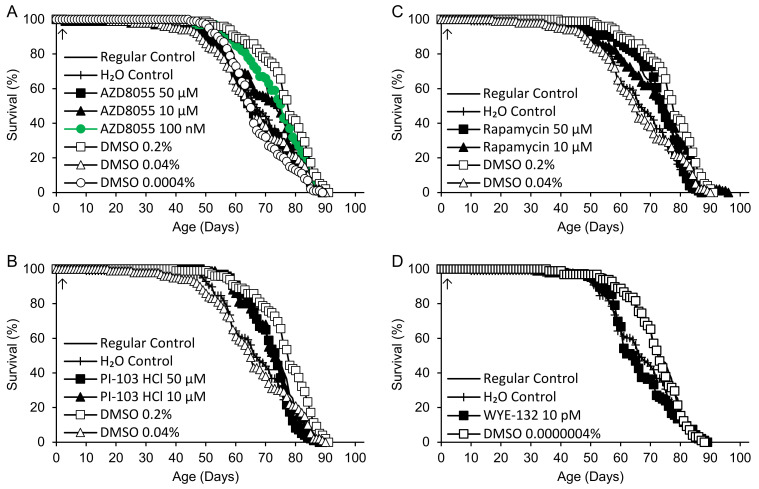
Life spans of flies supplemented with (**A**) AZD8055, (**B**) PI-103 hydrochloride (HCl), (**C**) rapamycin, (**D**) WYE-132 or DMSO control beginning at 2 days of adult life (arrows). Green: life extension vs. regular, H_2_O and DMSO controls (*p* < 0.050 based on logrank tests).

**Figure 4 ijms-25-11504-f004:**
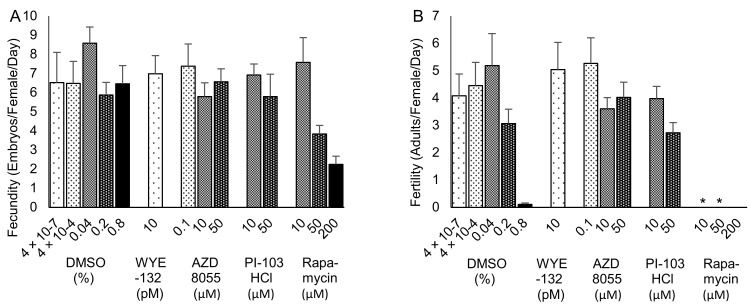
Fecundity (**A**) and fertility (**B**) of flies supplemented with AZD8055, PI-103 hydrochloride (HCl), rapamycin, WYE-132 or DMSO control. Results (mean ± SEM) are the number of eggs laid per female per day during the first four days of adulthood (**A**) and the corresponding number of adults developed from those eggs within 14 days at 25 °C (**B**). * *p* < 0.017.

**Figure 5 ijms-25-11504-f005:**
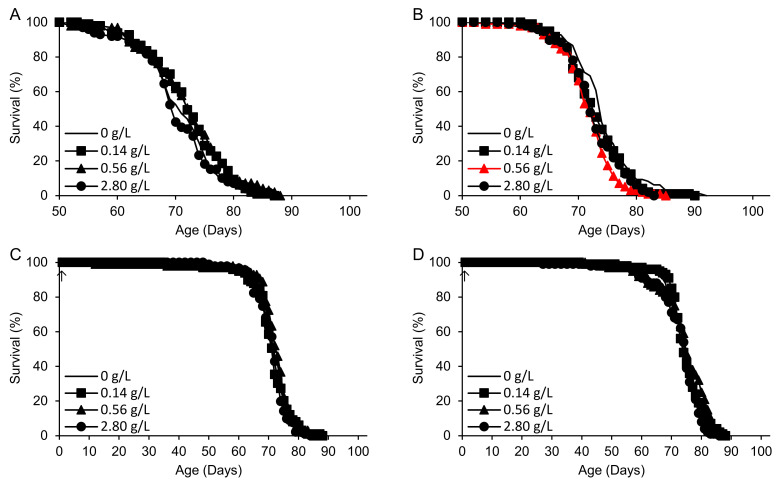
Survival times of flies supplemented with 2-HP-β-CD beginning at either 50 days (**A**,**B**) or 1 day of adult life (arrows) (**C**,**D**). (**A**–**C**) The standard medium. (**D**) A high-yeast/high-sugar medium. Red: life shortening vs. H_2_O control (*p* < 0.017 based on logrank tests with Bonferroni correction for three comparisons).

**Table 1 ijms-25-11504-t001:** IC_50_ of TOR/PI3 kinase inhibitors ^1^.

Inhibitor	TORC1 (nM) ^2^		TORC2 (nM) ^3^	PI3Kinase (nM) ^4^	References
AZD8055	0.13 ± 0.05 ^5^			3200–18,900 ^6^	[39]
Ku-0063794	~10		~10	>10,000	[50]
LY294002	1500 ^5^	~5000 ^7^		1400	[43,44,58]
PI-103 hydrochloride ^8^	20		83	3–250 ^6^	[56,57]
PP242	30 ^9^		58	102–2200 ^6^	[40]
PX-866-17OH	^10^			14–57 ^6,10^	[54]
Rapamycin	2 ^5,11^		—	—	[58]
Torin2	0.25 ^12^			200 ^12^	[36]
Wortmannin	200 ^5^	~200 ^7^, 300 ^13^		0.3–4	[44,47,48,53,58]
WYE-28	0.22 ± 0.06 ^5^			4271	[51]
WYE-132	0.19 ± 0.07 ^5^		^3^	1179–>10,000 ^6^	[52]

^1^ = not an inhibitor of the listed kinase. Blank spaces indicate no known test data. Values may not be directly comparable between inhibitors in cases where different substrates were used for IC_50_ determinations. ^2^ Substrates were biotinylated S6K for AZD8055; S6K1 T389 for Ku-0063794 and Torin2; His6-S6K T389 for LY294002, rapamycin, wortmannin, WYE-28 and WYE-132, except as noted in footnotes below; and 4E-BP1 for PI-103 and PP242. ^3^ Substrates were Akt for PI-103; Akt S473 for Ku-0063794 and WYE-132 (IC_50_ not stated for WYE-132 but comparable to TORC1 IC_50_ based on immunoblot [52]); and mTOR labeling in the presence of PKB for PP242. ^4^ Substrates were PIP_2_ for AZD8055, Ku-0063794, WYE-28 and WYE-132; phosphatidylinositol for LY294002 [43], PI-103, PP242 [40] and wortmannin [47,48]; and Akt T308 for Torin2. The substrate was not specified for PX-866-17OH. ^5^ Values based on truncated, FLAG-tagged mTOR. For AZD8055, full-length mTOR IC_50_ = 0.8 ± 0.2 nM with the 4E-BP-1 substrate. Values for AZD8055 also varied with ATP concentration, up to ~15 nM at 200 µM ATP. ^6^ Multiple values represent tests in >1 PI3K class I isoform (PI3Kα, β, γ or δ). ^7^ Values reported for TOR autophosphorylation without differentiating between TORC1 and TORC2 [44]. ^8^ Values for PI-103. ^9^ IC_50_ = 8 nM for mTOR (Invitrogen). ^10^ For the parent compound PX-866, TOR IC_50_ >10,000 nM, PI3K IC_50_ = 0.5 nM [53] or PI3K IC_50_ = 39–183 nM [54]; assay methods not specified. ^11^ IC_50_ = 0.2 nM for the inhibition of neutrophil S6K enzymatic activity, not linked conclusively to TOR [59]. ^12^ Cellular EC_50_ values. In vitro mTOR IC_50_ = 2.1 nM with the substrate GFP-4E-BP1. ^13^ Method not specified [53].

**Table 2 ijms-25-11504-t002:** Mean survival times of flies supplemented with DMSO, PP242 or DFP beginning at 64 days of adult life.

Supplement (Concentration) ^1^	*n* ^2^	*p* ^3^	Survival Time (Days)	% vs. H_2_O (*p*)		% vs. Regular (*p*)	
Control (Regular)	201 (3)		21.0				
Control (H_2_O)	90 (1)		17.4			−17.0	(0.002)
DMSO		<0.0005					
DMSO (2%)	86 (7)		4.4	−74.7	(<0.0005)	−79.0	(<0.0005)
DMSO (0.2%)	95 (0)		16.1	−7.4	(0.92)	−23.1	
DMSO (0.02%)	91 (4)		19.8	+13.7	(0.055)	−5.6	
DMSO (0.002%)	89 (0)		18.1	+3.8	(0.21)	−13.8	
DMSO (0.0002%)	84 (4)		18.5	+6.3	(0.79)	−11.8	
PP242		<0.0005					
PP242 (649 nM)	88 (1)		19.1	+9.7	(0.048)	−8.9	(0.60)
PP242 (64.9 nM)	86 (0)		18.5	+6.0	(0.15)	−12.0	
PP242 (21.6 nM)	88 (4)		18.2	+4.3	(0.63)	−13.4	
PP242 (6.49 nM)	95 (2)		15.0	−13.7	(0.030)	−28.3	(<0.0005)
PP242 (649 pM)	88 (0)		15.9	−8.8	(0.028)	−24.3	(<0.0005)
DFP		0.003					
DFP (719 µM)	85 (6)		16.2	−6.9	(0.45)	−22.7	
DFP (71.9 µM)	64 (1)		15.9	−8.9	(0.99)	−24.3	
DFP (7.19 µM)	90 (0)		18.6	+6.4	(0.19)	−11.6	
DFP (719 nM)	87 (0)		20.9	+19.6	(0.012)	−0.7	(0.98)
DFP (71.9 nM)	91 (1)		20.2	+16.0	(0.006)	−3.7	(0.82)

^1^ Concentrations are final values after diluting 100 µL of the supplement in 4.9 mL of the fly medium. ^2^ *n* = number of flies with recorded ages at death (censored data: numbers of additional flies that escaped or were accidentally killed at a known age are shown in parentheses and include flies lost due to excess moisture on the food surface on the first day of supplementation). Totals do not add to 100 (or 225 for the regular control group) because of deaths or escapes prior to 64 d, losses at unknown times and one vial being removed from the DFP 71.9 µM group due to experimental error. ^3^ *p* values for each supplement are for logrank tests including the H_2_O control group and all concentrations of the supplement. Where *p* < 0.050, tests were performed for each dose individually. Where *p* < 0.050 again, comparisons were also made with the regular control group.

**Table 3 ijms-25-11504-t003:** Adult mean life spans of flies supplemented with DMSO, spermidine or TOR/PI3K inhibitors beginning in early adult life.

Supplement (Concentration) ^1^	*n* ^2^	*p* ^3^	Life Span (Days)	% vs. H_2_O (*p*)		% vs. Regular (*p*)	
Control (Regular)	101 (0)		73.9				
Control (H_2_O)	95 (3)		74.4			0.6	
Spermidine		<0.0005					
Spermidine (10 mM)	98 (0)	<0.0005	66.2	−10.9	(<0.0005)	−10.4	(<0.0005)
Spermidine (1 mM)	96 (2)	0.87	73.6	−1.1		−0.5	
Spermidine (100 µM)	90 (1)	0.98	74.3	−0.1		+0.5	
Spermidine (10 µM)	88 (4)	0.98	73.5	−1.1		−0.6	
DMSO		0.88					
DMSO (0.2%)	96 (0)		73.3	−1.4		−0.9	
DMSO (0.1%)	93 (3)		73.2	−1.5		−1.0	
DMSO (0.04%)	94 (3)		73.0	−1.8		−1.3	
DMSO (0.02%)	94 (2)		72.0	−3.2		−2.6	
WYE-28		0.11					
WYE-28 (1 µM)	98 (2)		74.6	+0.4		+0.9	
WYE-28 (100 nM)	73 (2)		72.2	−2.9		−2.3	
WYE-28 (10 nM)	99 (0)		74.0	−0.5		+0.1	
WYE-28 (1 nM)	72 (2)		71.8	−3.4		−2.8	
WYE-28 (100 pM)	95 (3)		74.2	−0.2		+0.4	
WYE-28 (10 pM)	94 (1)		76.6	+3.1		+3.6	

Control (Regular)	94 (1)		76.2				
Control (H_2_O)	98 (0)		75.7			−0.6	
WYE-132		<0.0005					
WYE-132 (1 µM)	91 (8)	0.083	73.5	−3.0		−3.6	
WYE-132 (100 nM)	90 (2)	0.53	76.1	+0.4		−0.2	
WYE-132 (10 nM)	98 (0)	0.57	76.6	+1.1		+0.5	
WYE-132 (1 nM)	97 (1)	0.60	76.5	+1.0		+0.4	
WYE-132 (100 pM)	94 (4)	0.31	77.3	+2.0		+1.4	
WYE-132 (10 pM)	95 (4)	0.001	80.3	+6.0	(<0.0005)	+5.3	(0.012)
LY294002		0.12					
LY294002 (10 µM)	94 (0)		75.6	−0.2		−0.8	
LY294002 (5 µM)	93 (1)		75.3	−0.5		−1.1	
LY294002 (1 µM)	96 (0)		76.9	+1.5		+0.9	
LY294002 (100 nM)	99 (0)		74.2	−2.0		−2.6	

Control (Regular)	96 (0)		73.3				
Control (H_2_O)	95 (2)		75.3			+2.7	
AZD8055		0.036					
AZD8055 (10 µM)	98 (2)	0.009	78.1	+3.7	(0.006)	+6.6	(0.006)
AZD8055 (1 µM)	89 (0)	0.29	76.0	+0.9		+3.7	
AZD8055 (100 nM)	107 (3)	0.013	78.1	+3.7	(0.011)	+6.5	(0.008)
AZD8055 (10 nM)	80 (1)	0.38	75.4	+0.1		+2.8	
AZD8055 (1 nM)	105 (3)	0.078	76.7	+1.8		+4.6	
AZD8055 (100 pM)	90 (1)	0.45	75.1	−0.3		+2.4	
Rapamycin		<0.0005					
Rapamycin (400 µM)	90 (4)	<0.0005	41.8	−44.5	(<0.0005)	−42.9	(<0.0005)
Rapamycin (200 µM)	99 (0)	<0.0005	47.7	−36.7	(<0.0005)	−34.9	(<0.0005)
Rapamycin (100 µM)	92 (5)	<0.0005	54.2	−28.0	(<0.0005)	−26.1	(<0.0005)

Control (Regular)	98 (1)		73.4				
Control (H_2_O)	97 (2)		75.7			+3.2	
Rapamycin		0.22					
Rapamycin (10 µM)	96 (0)		74.2	−2.0		+1.1	
Rapamycin (1 µM)	96 (2)		72.5	−4.3		−1.3	
Rapamycin (500 nM)	91 (2)		73.1	−3.4		−0.3	
Rapamycin (100 nM)	94 (2)		73.5	−3.0		+0.1	
Ku-0063794		0.062					
Ku-0063794 (10 µM)	99 (3)		75.8	+0.1		+3.3	
Ku-0063794 (1 µM)	94 (0)		77.2	+1.9		+5.1	
Ku-0063794 (100 nM)	100 (0)		75.6	−0.1		+3.0	
Ku-0063794 (10 nM)	105 (1)		77.3	+2.1		+5.3	
Ku-0063794 (1 nM)	71 (1)		74.7	−1.4		+1.8	
Wortmannin		<0.0005					
Wortmannin (10 µM)	99 (0)	<0.0005	22.6	−70.2	(<0.0005)	−69.2	(<0.0005)
Wortmannin (1 µM)	94 (2)	0.21	72.6	−4.1		−1.0	
Wortmannin (500 nM)	96 (2)	0.27	74.7	−1.4		+1.8	
Wortmannin (100 nM)	98 (1)	0.11	73.9	−2.4		+0.7	
Wortmannin (10 nM)	98 (1)	0.11	74.4	−1.7		+1.4	
Wortmannin (1 nM)	99 (1)	0.092	76.2	+0.7		+3.8	

Control (Regular)	67 (4)		75.5				
Control (H_2_O)	99 (2)		75.4			−0.1	
PX-866-17OH		0.26					
PX-866-17OH (10 µM)	99 (2)		78.0	+3.5		+3.4	
PX-866-17OH (1 µM)	100 (0)		74.2	−1.6		−1.7	
PX-866-17OH (500 nM)	98 (1)		76.7	+1.7		+1.6	
PX-866-17OH (100 nM)	93 (3)		75.6	+0.2		+0.1	
PX-866-17OH (10 nM)	97 (2)		76.7	+1.7		+1.6	
PX-866-17OH (1 nM)	94 (3)		74.5	−1.2		−1.3	

Control (H_2_O)	91 (4)		75.4				
PI-103 HCl		0.029					
PI-103 HCl (10 µM)	94 (1)	0.010	78.4	+4.0			
PI-103 HCl (1 µM)	98 (2)	0.95	75.2	−0.3			
PI-103 HCl (100 nM)	96 (3)	0.41	76.6	+1.5			
PI-103 HCl (10 nM)	97 (1)	0.36	76.4	+1.3			
PI-103 HCl (1 nM)	94 (2)	0.91	75.6	+0.3			
PI-103 HCl (100 pM)	98 (2)	0.75	74.4	−1.3			
Torin2		0.40					
Torin2 (10 µM)	93 (4)		75.3	−0.1			
Torin2 (1 µM)	98 (0)		76.3	+1.2			
Torin2 (100 nM)	97 (2)		75.4	−0.1			
Torin2 (10 nM)	99 (1)		74.9	−0.7			
Torin2 (1 nM)	96 (3)		75.0	−0.6			
Torin2 (100 pM)	97 (3)		75.3	−0.1			

^1^ Concentrations are final values after diluting 100 µL of the supplement in 4.9 mL of the fly medium. ^2^ *n* = number of flies with recorded ages at death (censored data: numbers of additional flies that escaped or were accidentally killed at a known age are shown in parentheses). Totals do not add to 100 because of losses at unknown times and one vial each being lost or removed from the WYE-28 100 nM and 1 nM groups, Ku-0063794 1 nM and the regular control group for PX-866-17OH due to experimental error. ^3^ *p* values for each supplement are for logrank tests including both control groups and all concentrations of the supplement. Where *p* < 0.050, tests were performed for each dose and both control groups. Where *p* < 0.050 again, comparisons were also made for each control group separately.

**Table 4 ijms-25-11504-t004:** Adult mean life spans of flies supplemented with AZD8055, PI-103 hydrochloride (HCl), rapamycin (Rapa) or WYE-132 and DMSO control groups—replication experiment.

Supplement (Concentration) ^1^	*n* ^2^	*p* ^3^	Life Span (Days)	% vs. H_2_O (*p*)		% vs. Regular (*p*)		% vs. DMSO (*p*) ^4^	
Control (Regular)	98 (0)		72.5						
Control (H_2_O)	98 (0)		66.8			−7.8	(0.065)		
DMSO		<0.0005							
DMSO (0.2%)	99 (0)	<0.0005	76.5	+14.5	(<0.0005)	+5.6	(<0.0005)		
DMSO (0.04%)	100 (0)	0.20	65.6	−1.9		−9.5			
DMSO (0.0004%)	100 (0)	0.022	66.6	−0.3	(0.45)	−8.1	(0.005)		
DMSO (0.0000004%)	98 (1)	0.12	71.8	+7.4		−1.0			
AZD8055		0.001							
AZD8055 (50 µM)	101 (0)	0.16	66.6	−0.4		−8.1		−13.0	
AZD8055 (10 µM)	100 (0)	0.030	69.9	+4.6	(0.020)	−3.6	(0.39)	+6.6	(0.12)
AZD8055 (100 nM)	100 (0)	0.001	73.4	+9.8	(<0.0005)	+1.3	(0.030)	+10.2	(<0.0005)
PI-103 HCl		0.10							
PI-103 HCl (50 µM)	97 (1)		71.8	+7.5		−0.9		−6.1	
PI-103 HCl (10 µM)	100 (0)		72.2	+8.1		−0.3		+10.2	
Rapamycin		0.003							
Rapamycin (50 µM)	99 (1)	0.12	72.2	+8.0		−0.4		−5.6	
Rapamycin (10 µM)	102 (2)	0.002	71.4	+6.9	(0.002)	−1.4	(0.069)	+9.0	(0.007)
WYE-132 (10 pM)	96 (3)	0.033	66.0	−1.2	(0.59)	−8.9	(0.008)	−8.0	(0.016)

^1^ Concentrations are final values after diluting 100 µL of the supplement in 4.9 mL of the fly medium. ^2^ *n* = number of flies with recorded ages at death (censored data: numbers of additional flies that escaped or were accidentally killed at a known age are shown in parentheses). Totals do not add to 100 because of losses at unknown times. ^3^ *p* values for each supplement are for logrank tests including regular and H_2_O control groups and all concentrations of the supplement. Where *p* < 0.050, tests were performed for each dose and both control groups. Where *p* < 0.050 again, comparisons were made for each control group separately. ^4^ Based on 50 µM supplement vs. 0.2% DMSO, 10 µM supplement vs. 0.04% DMSO, 100 nM supplement vs. 0.0004% DMSO, 10 pM supplement vs. 0.0000004% DMSO.

**Table 5 ijms-25-11504-t005:** Survival times of flies supplemented with 2-HP-β-CD beginning at 50 days and life spans of flies supplemented with 2-HP-β-CD beginning at 1 day of adult life.

Supplement (Concentration) ^1^	*n* ^2^	*p* ^3^	Survival Time (Days)	% vs. H_2_O (*p*)		Medium (Figure)	Supplement Onset (Days)
Control (H_2_O)	97 (0)	0.079	20.8			Standard	50
2-HP-β-CD (0.14 g/L)	97 (0)		22.1	+6.5		(Figure 5A)	
2-HP-β-CD (0.56 g/L)	97 (0)		22.3	+7.1			
2-HP-β-CD (2.80 g/L)	98 (0)		20.5	−1.6			
Control (H_2_O)	97 (0)	0.002	24.2			Standard	50
2-HP-β-CD (0.14 g/L)	97 (1)		23.1	−4.8	(0.12)	(Figure 5B)	
2-HP-β-CD (0.56 g/L)	98 (0)		21.8	−10.2	(<0.0005)		
2-HP-β-CD (2.80 g/L)	96 (0)		22.6	−6.6	(0.017)		
Control (H_2_O)	100 (0)	0.31	71.1			Standard	1
2-HP-β-CD (0.14 g/L)	99 (0)		70.9	−0.2		(Figure 5C)	
2-HP-β-CD (0.56 g/L)	98 (0)		72.0	+1.3			
2-HP-β-CD (2.80 g/L)	91 (0)		71.0	−0.0			
Control (H_2_O)	100 (0)	0.040	71.0			High	1
2-HP-β-CD (0.14 g/L)	100 (0)		74.6	+5.1	(0.27)	Sugar/	
2-HP-β-CD (0.56 g/L)	100 (0)		74.3	+4.6	(0.032)	Yeast	
2-HP-β-CD (2.80 g/L)	100 (0)		72.7	+2.3	(0.65)	(Figure 5D)	

^1^ Concentrations are final values after diluting 1 mL of the supplement in 99 mL of the fly medium. ^2^ *n* = number of flies with recorded ages at death (censored data: numbers of additional flies that escaped or were accidentally killed at a known age are shown in parentheses). Totals do not add to 100 because of deaths or escapes prior to the onset of supplementation and losses at unknown times. ^3^ *p* values for each experiment are for logrank tests including the H_2_O control group and all concentrations of 2-HP-β-CD. Where *p* < 0.050, tests were performed for each dose vs. the H_2_O control group. The Bonferroni-adjusted significance threshold for multiple comparisons was *p* < 0.017.

## Data Availability

The raw data supporting the conclusions of this article will be made available by the authors on request.

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
