# Peer review of "Effects of Target of Rapamycin and Phosphatidylinositol 3-Kinase Inhibitors and Other Autophagy-Related Supplements on Life Span in y w Male Drosophila melanogaster"

_ijms, 2024, doi:10.3390/ijms252111504_

Round 1

Reviewer 1 Report

Comments and Suggestions for Authors

The authors examined the effects of several pharmacological molecules targeting the mTOR and PI3K/Akt pathways on the survival of male Drosophila. While the title suggests the study aims to investigate the impact of these molecules on longevity, many experiments instead focus on survival rate, which is not directly linked to lifespan. For instance, it is widely accepted that mTOR inhibition can extend lifespan to varying degrees. However, in this manuscript, mTOR inhibition led to a significant decrease in survival rate. The authors speculate that this might be due to the high dosage used, yet they indicate that results from lower dosages will be presented in a separate manuscript. This approach is problematic. The rapamycin experiments, including the full dosage range, should be presented within this manuscript as they are crucial for validating the experimental design and logic throughout the study.

Another major issue is the lack of statistical information. The number of individuals used to generate the survival rate curves should be provided. Additionally, despite the authors claiming a slight increase in lifespan in the wortmannin experiments, no statistical test information was included to support this claim (especially as the effect is rather trivial).

In line with this, the experiments of Despite that this is already very confusing, the design of the experiments, especially the dosage choice, is not clearly stated. I found this very confusing. It's been shown in many contexts that manipulating mTOR pathway activity can influence the lifespan and that inhibition of mTOR and specifically, the nhibition of mTOR can extend the lifespan by various extent depending on the contexts. However here the authors show that inhibition of mTOR dramatically decrease the survival rate of drosophila individuals. In addition, multiple reports indicate a link between ageing and PI3K/Akt pathway activities.  Here the authors use the drosophila system to test the effect of 

Author Response

Comments 1: The authors examined the effects of several pharmacological molecules targeting the mTOR and PI3K/Akt pathways on the survival of male Drosophila. While the title suggests the study aims to investigate the impact of these molecules on longevity, many experiments instead focus on survival rate, which is not directly linked to lifespan.

Response 1: We removed the word “longevity” but replaced it with “life span” later in the title (p. 1, lines 3-4). The majority of the experiments feature supplementation throughout adult life, so we use “life span” as a column title in Tables 3 and 4. We added “time” after “survival” in the column headings of Tables 2 and 5 to match their titles (p. 5, line 202 and p. 14, line 333) and to make clear that the times are again spans of time, but only after supplementation began. Thus, the mean life span can be derived by adding the age of onset of supplementation to the survival times. None of the data presented are survival rates, i.e. proportions of flies surviving to a fixed time point.

Comments 2: For instance, it is widely accepted that mTOR inhibition can extend lifespan to varying degrees. However, in this manuscript, mTOR inhibition led to a significant decrease in survival rate. The authors speculate that this might be due to the high dosage used, yet they indicate that results from lower dosages will be presented in a separate manuscript. This approach is problematic. The rapamycin experiments, including the full dosage range, should be presented within this manuscript as they are crucial for validating the experimental design and logic throughout the study.

Response 2: To clarify, the full dosage range for rapamycin is presented in Table 3. The unpublished experiments mentioned in the Discussion (paragraph 1, lines 352-354) test the effects of a high dose (200 µM) on life span in other strains or environmental conditions that were not used in this study. We originally intended to incorporate those experiments within this manuscript, but they expanded to such an extent that adding them here would require another few months to complete the analysis, add another four authors and increase the current manuscript by 50-100%, which would be excessive given its current length of >22 pages.

Comments 3: Another major issue is the lack of statistical information. The number of individuals used to generate the survival rate curves should be provided.

Response 3: Thank you for this request. We had placed a blanket statement in section 4.3 that each test began with four groups of 25 flies, but the numbers escaping or lost to attrition varied and in a few cases an entire vial had to be withdrawn due to mass escapes. Therefore, we have added a column each to Tables 2-5 showing the numbers of dying flies used to generate the graphs for every group as well as the numbers of flies with censored data (due to escapes or occasional flies being killed during the daily transfer to new vials). We also added explanatory footnotes to those tables.

While checking the data to make these revisions, we uncovered a small error in data entry leading to the deaths of three flies in the WYE-28 100 nM group at age 66 days being attributed to the H2O control group. We therefore replaced the affected figure panels (Figure 2A, B, L), although the changes are imperceptible, and repeated all of the statistical analyses for those groups, leading to incremental changes of one number in the text (p. 6, paragraph 1, line 223) and the mean life spans and percentages calculated for those groups in Table 3 (p. 9, line 252). All changes are shown in red typeface. These changes did not cause any groups to rise or fall across the threshold of statistical significance and did not alter the interpretation of the results.

Comments 4: Additionally, despite the authors claiming a slight increase in lifespan in the wortmannin experiments, no statistical test information was included to support this claim (especially as the effect is rather trivial).

Response 4: Regarding wortmannin increasing life span, we found a life-shortening effect, but only at the highest dose. Might the reviewer be referencing the 100 nM AZD8055 group? We agree that the difference is trivial, but for completeness we thought it should be mentioned in the Abstract. As the reviewer notes, others have found that mTOR inhibition is beneficial and our results mostly do not fit that theme, so we did not want to overstate the case by ignoring the exception to our general conclusion.

Regarding statistical testing, we performed logrank testing for all groups as described in section 4.5 (p. 19, lines 547-568) and ANOVA for fertility (p. 19, lines 569-577). The p values from the logrank tests have columns in Tables 2-5. We did not restate them in the text, but have now added “statistically significant” the first time a life span comparison was made (p. 4, paragraph 1, lines 185-186) and a text citation to the p values in Table 2 (p. 4, paragraph 1, lines 188-189). The logrank test results in Table 3 are already cited on p. 6, line 220.

Comments 5: In line with this, the experiments of Despite that this is already very confusing, the design of the experiments, especially the dosage choice, is not clearly stated. I found this very confusing.

Response 5: The doses were selected to cover the widest practical range, typically several orders of magnitude, including the range tested by others in Drosophila where possible and ranging as high as practical given the price of the supplements and their solubility, and as low as would be practical given the IC50s of compounds applied directly in vitro. We revised the text (pp. 3-4, lines 149-152) to make this clearer.

Comment 6: It's been shown in many contexts that manipulating mTOR pathway activity can influence the lifespan and that inhibition of mTOR and specifically, the nhibition of mTOR can extend the lifespan by various extent depending on the contexts. However here the authors show that inhibition of mTOR dramatically decrease the survival rate of drosophila individuals. In addition, multiple reports indicate a link between ageing and PI3K/Akt pathway activities.  Here the authors use the drosophila system to test the effect of 

Response 6: This comment appears to have been truncated, but in response to the portion we received, we agree that inhibiting mTOR has sometimes been shown to extend lifespan and that it depends on context. The existing literature that we cited is in fact quite mixed regarding mTOR inhibition and life span in Drosophila. There are positive results, but there are several negative results reported by others too, and some of the positive effects are quite slight in magnitude. Altogether, with the exception of rapamycin, these inhibitors have been studied on a rather piecemeal basis. Our study adds to the set of conditions under which inhibition has mainly neutral or negative effects. We tried to be more comprehensive in the range of compounds and doses that we tested, but we only used one fly strain with one diet/environment, so there is still more work to be done. We also restricted our conclusion about the lack of benefit of TOR/PI3K inhibition to this strain of flies at the end of the Discussion. In response to the reviewer’s comment, we changed our previous statement that it “might” be beneficial in other strains/environments to “can be beneficial” (p. 17, paragraph 3, line 479).

Reviewer 2 Report

Comments and Suggestions for Authors

This is a fairly interesting study by Bearden and co-workers, although not highly original. It aimed to test a wide range of mTOR and PI3K inhibitors on a strain of male D Melanogaster flies for longevity study.

The study is well written and clear.  Is a bit pity that the survival curves are so close and overlapping in many graphs. One may wonder if it would have been possible to cut the graphs at 30 days to allow for higher resolution of day 50-90. Although they data is also in tables.

None of the compounds had major life prolonging effect.

The study is mostly descriptive and in some parts it more resembles a tox study, using very high or extremely high concentrations of compounds that has no relevance whatsoever for the study of longevity in any species (though they serve as internal control for compound stability).

The study relies on testing several compounds and that can be a problem in controlling since it increases the logistical issues and can lead to errors.

The rational to use some of the compounds is not fully clear, a few of them are likely to have extensive poly-pharmacology and inhibit many kinases in the cells. For example, wortmannin, PP242 and Torin2 are at mid or high concentrations highly toxic to normal and mammalian cancer cell lines cultured in vitro. Hence, it is notable that they have surprisingly much less toxic effect on longevity in flies than expected (suggesting impaired uptake). With wortmannin for example we are at 10um (fig 2), while extrapolating from mammalian cells in vitro you would expect such effect to be around 1- or 0.1uM.

The study discussion brings up a many critical issues and limitations of the study which is good (and necessary).

Minor points of critique that the authors may want to further address in discussion and data presentation:

1.        Rapamycin is used at extremely high concentrations in experiments, even in the low concentration we have 100nM. That is notable, and it needs to be said that rapamycin is a natural compound, and it could be that there could be targets in the cells that have not yet been identified. And I think that some statements that the authors do that relies on using this high concentration, is unnecessarily critical. Looking at data in figure 4 for example, 200uM Rapa for fertility. This is interesting but has very little relevance and is misleading in the context of the main hypothesis in the study.

2.        Spermidine uptake is not at all clear, and while there is some effect at 10mM (figure 2), it is unclear what this effect is coming from. Could as well be some indirect buffering effect due to high conc of spermidine. I don’t see the physiological relevance of this concentration and way of testing spermidine in relation to longevity.

3.        High concentrations of some cpds used while no major negative effect seen on viability suggest uptake/ exposure issues.

Author Response

Comments 1: This is a fairly interesting study by Bearden and co-workers, although not highly original. It aimed to test a wide range of mTOR and PI3K inhibitors on a strain of male D Melanogaster flies for longevity study.

The study is well written and clear.  Is a bit pity that the survival curves are so close and overlapping in many graphs. One may wonder if it would have been possible to cut the graphs at 30 days to allow for higher resolution of day 50-90. Although they data is also in tables.

None of the compounds had major life prolonging effect.

Response 1: Thank you. We agree that there is substantial overlap. It seems best to include all of the results in the figures. The period from 0-30 days shows limited early mortality, which is an important indicator that the flies in the experiment were not already sick or injured at the outset. Perhaps we could include fewer doses per panel, but that would multiply the number of figure panels, which is already large. The overlap also reflects the lack of a major effect of most doses of most compounds. In cases of a large effect, as with the highest doses of wortmannin and rapamycin, the separation is very clear. We also used red/green coloration to highlight all of the significant differences, which serves to show that they exist but also that they are minor in most cases.

Comments 2: The study is mostly descriptive and in some parts it more resembles a tox study, using very high or extremely high concentrations of compounds that has no relevance whatsoever for the study of longevity in any species (though they serve as internal control for compound stability).

The study relies on testing several compounds and that can be a problem in controlling since it increases the logistical issues and can lead to errors.

Response 2: Yes, we reached the limits of our capacity in the number of compounds and doses tested in the experiment for Figure 2, Table 3. The numbers of escaped or missing (or excess) flies (included in response to reviewer 1) was a bit higher there than for Tables 4 and 5, but still low as a proportion. The most anomalous numbers were for AZD8055, where the replicate experiment in Table 4 yielded a consistent outcome with no attrition. Regarding the doses, we extended up to high doses because we did not know how much would be absorbed. We made additions to the Discussion to address the reviewer’s point here about toxicity (p. 16, paragraph 3, lines 422-426 and 431-435) and below about bioavailability (p. 16, paragraph 2, lines 415-421).

Comments 3: The rational to use some of the compounds is not fully clear, a few of them are likely to have extensive poly-pharmacology and inhibit many kinases in the cells. For example, wortmannin, PP242 and Torin2 are at mid or high concentrations highly toxic to normal and mammalian cancer cell lines cultured in vitro. Hence, it is notable that they have surprisingly much less toxic effect on longevity in flies than expected (suggesting impaired uptake). With wortmannin for example we are at 10um (fig 2), while extrapolating from mammalian cells in vitro you would expect such effect to be around 1- or 0.1uM.

Response 3: For these three compounds, a large part of our rationale was simply that others have already tested them, albeit in some cases not under conditions that led to the full-length control life spans that would give a basis to say that further life extension represented a slowing of aging. The remainder of the rationale is that we wanted to run multiple tests that inhibited different kinases to varying extents, expecting that some combination that was not necessarily foreseeable would have the greatest benefit. We have added this point to the Introduction, p. 3, paragraph 2, lines 132-135. If some combinations were beneficial and others were toxic, it would help to pinpoint how much of which particular kinases was responsible.

Comments 4: The study discussion brings up a many critical issues and limitations of the study which is good (and necessary).

Minor points of critique that the authors may want to further address in discussion and data presentation:

  1. Rapamycin is used at extremely high concentrations in experiments, even in the low concentration we have 100nM. That is notable, and it needs to be said that rapamycin is a natural compound, and it could be that there could be targets in the cells that have not yet been identified. And I think that some statements that the authors do that relies on using this high concentration, is unnecessarily critical. Looking at data in figure 4 for example, 200uM Rapa for fertility. This is interesting but has very little relevance and is misleading in the context of the main hypothesis in the study.

Response 4: We added the suggested statement about potential unidentified targets for rapamycin on p. 17, paragraph 2, lines 458-459. We also added to the Abstract that the dose of rapamycin affecting fertility was high (p. 1, line 26). It should be noted that the dose reaching the fly cells is likely not as high as the dose in the food.

Comments 5: 2.        Spermidine uptake is not at all clear, and while there is some effect at 10mM (figure 2), it is unclear what this effect is coming from. Could as well be some indirect buffering effect due to high conc of spermidine. I don’t see the physiological relevance of this concentration and way of testing spermidine in relation to longevity.

Response 5: We agree that for our study and most in the field looking at dietary supplementation of Drosophila, uptake is not easily documented. We at least looked into stability during storage before the experiment to narrow the scope of the problem, but it is an issue for the field as a whole. In this case, the dose range for spermidine was chosen to match what has been done by others [reference 5], but that reference included millimolar dosing for flies and mice in the diet while using nanomolar concentrations for isolated cells. We added this point to the Discussion (p. 16, paragraph 2, lines 415-417). This review is helping us to appreciate the difference of perspective between whole-animal and cell research regarding normal concentrations, which has not been a focus of much discussion in the literature.

Comments 6: 3.        High concentrations of some cpds used while no major negative effect seen on viability suggest uptake/ exposure issues.

Response 6: This point is beneficial to consider, as we have not worked with isolated cells and chemicals. We hope the additions to the Discussion (p. 16, paragraphs 2-3, lines 415-426, 431-435) address the reviewer’s concerns sufficiently.